# Urea as a Cocrystal Former—Study of 3 Urea Based Pharmaceutical Cocrystals

**DOI:** 10.3390/pharmaceutics13050671

**Published:** 2021-05-07

**Authors:** Fucheng Leng, Koen Robeyns, Tom Leyssens

**Affiliations:** Institute of Condensed Matter and Nanosciences, Université Catholique de Louvain, B-1348 Louvain-La-Neuve, Belgium; leng.fucheng@uclouvain.Be (F.L.); koen.robeyns@uclouvain.be (K.R.)

**Keywords:** cocrystal, urea, stability improvement, catechin, ellagic acid, 3-hydroxyl-2-naphthoic acid

## Abstract

Cocrystallization is commonly used for its ability to improve the physical properties of APIs, such as solubility, bioavailability, compressibility, etc. The pharmaceutical industry is particularly interested in those cocrystals comprising a GRAS former in connection with the target API. In this work, we focus on the potential of urea as a cocrystal former, identifying three novel pharmaceutical cocrystal systems with catechin, 3-hydroxyl-2-naphthoic and ellagic acid. Interestingly, the stability of catechin under high humidity or high temperature environment is improved upon cocrystallization with urea. Moreover, the solubility of ellagic acid is improved about 17 times. This work displays the latent possibility of urea in improving the physical property of drug molecules using a cocrystallization approach.

## 1. Introduction

Cocrystals have drawn increasing attention in recent years due to their ability to improve physical properties of active pharmaceutical ingredients (APIs) without changing the chemical structure of the original drug [1,2,3,4]. Although still in debate, a well-accepted definition describes cocrystals as “solids that are crystalline single-phase materials composed of two or more different molecules and/or ionic compounds generally in a stoichiometric ratio which are neither solvates nor simple salts” [5]. More specifically, pharmaceutical cocrystals combine a drug compound and a pharmaceutically acceptable coformer. There have been eight pharmaceutical cocrystals marketed up to date, with an even more important number undergoing clinical trials [6].

Indexed as a GRAS (General Regarded As Safe) compound, urea is an excellent choice of coformer from the pharmaceutical (safe) and economic (inexpensive) point of view. High water solubility coformers in general increase the solubility of the API when the cocrystal is formed [2,7,8]. Urea cocrystals are therefore expected to strongly impact the API solubility. Urea furthermore has functional groups frequently encountered in cocrystal hydrogen bonding patterns, and therefore forms an ideal candidate for co-crystal screening [9,10]. Various contributions already show the potential of urea for the improvement of physical properties compared to the original API [11,12,13,14]. Urea cocrystals raised the solubility of agomelatine 2.2 times [15]. Urea also improved the intrinsic dissolution rate of bumetanide [11], febuxostat [13] and niclosamide [14] in a variety of solvents.

We here present, three novel urea comprising pharmaceutical cocrystals with catechin, 3-hydroxyl-2-naphthoic acid and ellagic acid, all of which show interesting bioactivity. Specifically, ellagic acid is widely used in food and pharmaceutical industry owing to its antioxidant and anti-inflammatory effect [16,17]. The anti-diabetic effect of 3-hydroxyl-2-naphthoic acid has also been proved by previous reports [18]. Catechin is a flavanol which has been effectiveness as an antioxidant, and for improvement of the immune system response [19,20,21,22]. In this work we show how cocrystallization with urea, leads to a 17-fold solubility increase of ellagic acid, as well as an improvement of the physical stability of catechin. This work therefore further underlines the potential of urea for the improvement of physical properties of API through cocrystallization.

## 2. Materials and Methods

Materials. Catechin (98%) and 3-hydroxyl-2-naphthoic acid (98%) were bought from sigma-Aldrich, St. Louis, MO, USA. Ellagic acid (97%) was bought from Alfa Aesar, Haverhill, MA, USA. Urea was bought from Merck. Catechin hydrate is obtained by slurring catechin in water for 2 days, apart from that, all reagents were used as received.

Cocrystal screen. In a typical cocrystal screening experiment, 0.25 mmol urea and an equimolar amount of API are placed in an Eppendorf adding one stainless steel ball. After that, grinding was performed using a RETSCH Mixer Mill MM 400 with a beating frequency of 30 Hz for 90 min. Subsequently, the PXRD of the ground material is compared to that of the parent compounds. Upon apparition of novel peaks, grinding is performed under various ratios as well. When neat grinding did not lead to a full transformation, liquid assisted grinding was performed in parallel, adding 20 μL of solvent to the initial mixture of urea and target compound prior to grinding (solvents include methanol, ethanol, water, acetonitrile and isopropanol).

Mechanical synthesis of cocrystals. The urea:ellagic acid cocrystal (UE) can be obtained by liquid assisted grinding of 30 mg urea and 75 mg ellagic acid (2:1 molar ratio) using 20 μL of water or isopropanol. The urea:3-hydroxyl-2-naphthoic acid cocrystal (UH) as well as urea:catechin cocrystal (UC) can be obtained by dry grinding in a 1:1 molar ratio.

Single crystal growth. Methanol is added in a drop-wise manner to a vial containing 25 mg of catechin and 24 mg urea (1:5 molar ratio) until full dissolution is achieved. After that, the solution is left to evaporate. After one week, UC crystals are obtained of sufficient quality for SC-XRD. In a similar approach, single crystals of UH are obtained by evaporating an undersaturated methanol solution of urea and 3-hydroxyl-2-naphthoic acid (in a 1:3 molar ratio).

Powder X-ray diffraction and variable temperature X-ray powder diffraction *(VT-XRPD).* Powder X-ray diffraction of all samples are conducted on a Siemens D5000 diffractometer equipped with a Cu X-ray source operating at 40 KV and 40 mA (λ = 1.5418 Å) from 2 to 50 degree at the rate of 0.6 degree per minute. VT-XRPD of catechin hydrate is collected on a PANalytical X’Pert PRO automated diffractometer from 3 to 40 degree, equipped with an X’Celerator detector and an Anton Paar TTK 450 system for measurements at controlled temperature. Data were collected in open air in Bragg-Brentano geometry, using Cu-Kα radiation without a monochromator.

Structure Determination. Single crystal diffraction data for UC and UH were collected on a MAR345 image plate detector using Mo Kα radiation (λ = 0.71073 Å), generated by a Rigaku Ultra X18S rotating anode (Xenocs fox3d mirrors). For UC the crystal was flash frozen at 150K in a N_2_ flow prior to data collection. Data integration and reduction was performed by CrystAlisPro [23] and the implemented absorption correction was applied. Structure solution was performed by the dual-space algorithm in SHELXT [24] and the structure was further refined against *F*^2^ using SHELXL2014/7. All non-hydrogen atoms were refined anisotropically and hydrogen atoms were placed at calculated positions with temperature factors set at 1.2U_eq_ of the parent atoms (1.5U_eq_ for methyl and OH hydrogens).

For UE the structure was solved from powder diffraction measured on a STOE STADI P diffractometer using monochromated Cu Kα1 radiation in transmission mode (with the sample placed between zero scattering foils). Unit cell determination was performed by DICVOL and the structure was solved by DASH [25], the structure was subsequently optimized by Rietveld refinement in Fullprof [26]. The Rietveld profile is shown in Appendix A.

Thermogravimetric Analysis (*TGA*). Typically, the TGA analyses of all samples are performed from 30 to 450 °C using a heating rate of 5 °C/min with a continuous nitrogen flow of 50 mL/min, on a Mettler Toledo TGA/SDTA851e.

Differential Scanning Calorimetry (*DSC*). DSC measurements are performed on a TA DSC2500. Deposited in an aluminum Tzero pans with punctured hermetic lid, samples were heated from 20 °C up to 240 °C using a heating rate of 2 °C/min under a 50 mL/min continuous nitrogen flow.

Congruence experiments. Stoichiometric amounts of urea and API were added to 1 mL of solvent until dissolution no longer occurred and a suspension was obtained. After that, ground traces of cocrystal material were added to the suspension as seed material. After 3 days of slurrying at room temperature, the suspension was filtered and the solid analyzed by PXRD.

Solubility measurement. The solubility measurement is conducted in ethanol at room temperature. An excess amount of solid is added to 2 mL of ethanol and the suspension is left to slurry for 2 days reaching saturation. After that, the suspension is filtered, and the filtrate weighed and left for evaporation. Weighing the recovered solids, allows determining the amount of solvent as well as solid present in the filtrate, and hence the solubility.

## 3. Results

### 3.1. Cocrystal Screening

As our main goal was to show the potential of urea as a pharmaceutical cocrystal former, a screen involving 62 APIs was performed (Appendix A). Seven positive hits were identified in agreement with literature reported success rates of about 10% (Figure 1) [27]. From this data, APIs containing a phenol group have a higher likelihood of forming a cocrystal with urea. Four cocrystals were already reported in literature (Appendix A) (theophylline, nicotinamide, salicylic acid, and hydroquinone) [28,29,30]. We report here three new cocrystal systems with catechin, ellagic acid, and 3-hdyroxyl-2-naphthoic acid, which are discussed in detail.

### 3.2. Urea-Catechin Cocrystal (UC)

Urea and catechin cocrystallize in the monoclinic *P*2_1_ space group (Table 1). The unit cell contains two urea and two catechin molecules. As a hydrogen bond acceptor, the oxygen atom of each urea molecule is connected to a N–H group of a second urea molecule and to a phenolic hydroxyl of catechin. Furthermore, all hydroxyl groups are engaged in hydrogen bonds with hydroxyl groups of neighboring catechin molecules (Figure 2).

Figure 3 shows a PXRD overlay of the ground and starting materials (catechin is not displayed because the used catechin was amorphous), as well as the pattern simulated from the single crystal structure. As shown in this figure, the ground material matches the one from single crystal analysis, corresponding to the 1:1 cocrystal.

Urea shows a single melting point with onset at 134 °C immediately followed by a degradation as illustrated by the TGA analysis, similar to previous report [31]. The UC cocrystal shows a melting temperature of 176 °C with a corresponding heat of fusion of 162.78 J/g (Figure 4), which is followed by a degradation endotherm. Comparing the UC and the amorphous catechin material in terms of humidity stability, one notices the UC cocrystal to remain stable at 75% RH at 25 °C for a period of two weeks (Appendix A), whilst storing the amorphous material, leads to crystalline catechin hydrate under these conditions. Catechin hydrate in turn starts losing water at temperatures above 50 °C (Appendix A), transforming into the amorphous phase upon dehydration (Appendix A). Cocrystallization with urea, therefore, leads to a solid form of catechin which is much less moisture or thermo-sensitive.

### 3.3. Urea-3-Hydroxyl-2-Naphtoic Acid (UH)

Urea and 3-hydroxyl-2-naphthoic acid crystallize in the monoclinic *C*2/*c* space group in a 1:1 ratio. The carboxylic acid of 3-hydroxyl-2naphthoic acid, is connected to the amide group of urea through an amide-acid hetero-synthon. The phenyl hydroxyl forms an intramolecular hydrogen bond, as well as an intermolecular hydrogen bond with urea (Figure 5). Other hydrogen bonding patterns involve different urea molecules and are of the C = O–H–N type (Figure 5).

Figure 6 shows a PXRD overlay of the ground and starting materials, as well as the simulated pattern from the single crystal data. As shown in this figure, the ground material matches the single crystal phase, corresponding to a 1:1 cocrystal.

Further, 3-hydroxyl-2-naphthoic acid shows a single melting point with onset at 218 °C and an associated 173.3 J/g heat of fusion. The cocrystal in turn shows a single melting temperature at 155 °C with a heat of fusion 156.78 J/g followed by immediate degradation. As common for cocrystals, this melting point lies between that of both parent compounds. TGA confirms degradation upon melting for all phases (Figure 7).

### 3.4. Urea-Ellagic Acid (UE)

The UE cocrystal can be obtained by liquid assisted grinding of two equivalents of urea and one equivalent of ellagic acid using water (Figure 8). Grinding a 1:1 ratio, leads to cocrystal material with excess amount of ellagic acid. As attempts at growing a single crystal failed, the structure was resolved from the powder pattern. Urea and ellagic acid cocrystalize in the *P*-1 space group, with two urea and one ellagic acid molecule in the unit cell (Appendix A). Ellagic acid is found on a crystallographic inversion center. For ellagic acid, the oxygen atoms in the ester group of ellagic acid serve as hydrogen bond acceptor, connecting to amide groups from urea molecules. On the other hand, the phenolic hydroxyl groups in ellagic acid serve as hydrogen bond donor to the carbonyl oxygen of a urea molecule (Figure 9).

Thermal analysis of ellagic acid showed our initial powder to contain a mixture of the hydrate and anhydrate phase as shown in Figure 9. TGA of ellagic acid shows a mass loss of 2.5% at 103 °C, suggesting a quarter of ellagic acid used here is under the dihydrate form. DSC confirms this water loss. Ellagic acid has a reported melting temperature of 350 °C [32]. The co-crystal shows a single endotherm peak at 222 °C, corresponding to the melting point of the cocrystal. TGA shows melting to be followed by immediate degradation (Figure 10).

### 3.5. Solution Behavior

The solution behavior of the novel phases was evaluated in various solvents. Initially, the cocrystals were suspended in a solvent to evaluate their congruency. Congruency implies that stoichiometric amounts of the cocrystal components lead to the cocrystal as the only stable phase in suspension, while non-congruency means that one of the parent compounds crystallizes out (or a mixture of cocrystal and a parent compound). UH behaves congruently in ethanol, acetonitrile and isopropanol, whereas it is not congruent in water or methanol (Appendix A), with 3-hydroxyl-2-naphthoic acid crystallizing out. UE behaves congruently in methanol, ethanol, acetonitrile and isopropanol. In water, ellagic acid hydrate is obtained (Appendix A). UC crystalizes congruently in all organic solvents used here and incongruently in water, with catechin hydrate crystallizing out (Figure 11). In mixed water/methanol solvents, UC behaves congruently for solvent mixtures of 1:9 to 4:6 water/methanol ratios (Appendix A). When the water/methanol ratio varies from 5:5 to 6:4, a recently identified catechin methanol solvate-hydrate crystallizes out (catechin:water:methanol 2:2:1) (Appendix A) [33]. With an even higher water/methanol ratio, a PXRD profile different from any known form is obtained (Appendix A). Drying this phase under ambient conditions yields catechin hydrate, suggesting another solvate of catechin was likely obtained. Slurrying catechin on its own in water/methanol ratios from 7:3 to 9:1, only gives the catechin hydrate, which means urea likely plays a role in the stabilization of the yet unknown catechin solvate (Appendix A).

As all three new cocrystals behave congruently in ethanol, solubility measurements were conducted in this solvent. For UC and UH cocrystal, a solubility of 0.595 mol/L and 0.439 mol/L is obtained, which is lower than that of the parent compound (0.736 mol/L and 0.599 mol/L respectively). For ellagic acid, the behavior is inverted, with the solubility being raised from 0.52 mmol/L to 9.04 mmol/L, showing the potential of cocrystallization to strongly impact the solubility behavior of poorly soluble drugs. Solubility of a cocrystal depends on the free energy of the novel cocrystal as well as the solution free energy of dissolved compounds and their solution interaction. Predicting this solubility merely on the structure is not feasible. The increase in solubility for ellagic acid is not surprising as the solubility of ellagic acid is extremely low. Very likely a variation of free energy of the solid structure as well as a positive interaction between both components in solution needs to be taken into account.

## 4. Conclusions

In this work, three novel cocrystals involving urea were identified, targeting catechin, ellagic acid, and 3-hydroxyl-2-naphthoic acid. Urea is a GRAS compound that is a promising coformer with a potential strong impact on the solubility of the target compound, as shown here for a 18-fold solubility increase for ellagic acid. Furthermore, we showed how the stability of the target compounds can be impacted and improved upon by cocrystallization with urea.

## Figures and Tables

**Figure 1 pharmaceutics-13-00671-f001:**
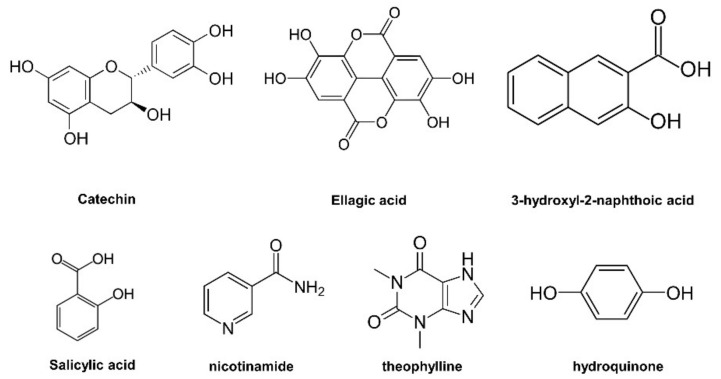
Chemical structure of the active pharmaceutical ingredients used in our screen which form cocrystals with urea.

**Figure 2 pharmaceutics-13-00671-f002:**
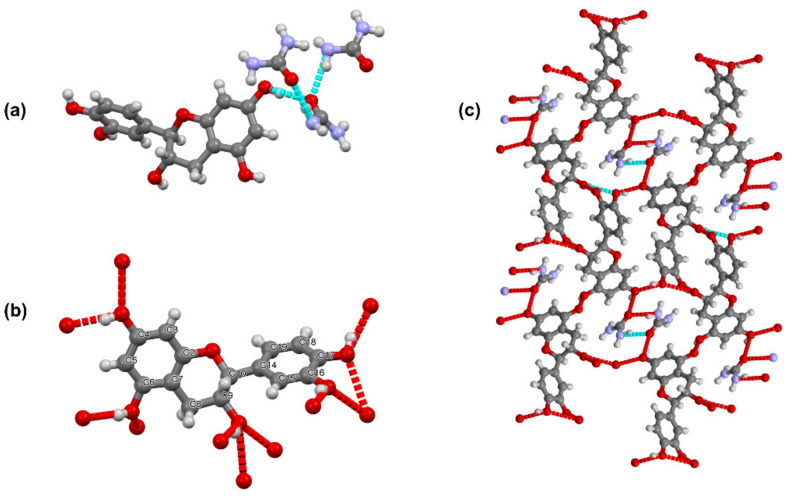
(**a**) Hydrogen bonding around a urea molecule in the UC cocrystal. (**b**) Hydrogen bonding around a catechin molecule in the cocrystal. (**c**) View along the a axis.

**Figure 3 pharmaceutics-13-00671-f003:**
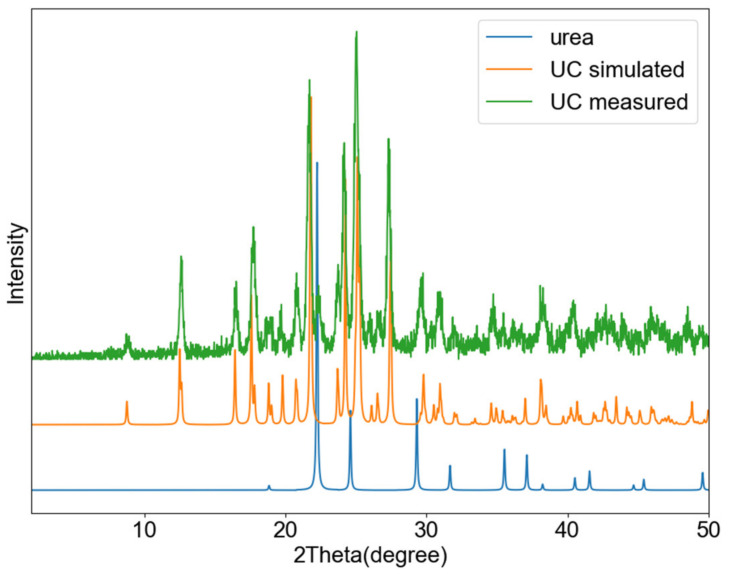
PXRD profiles of UC obtained by grinding (**green**), the simulated pattern of the UC cocrystal (**orange**), and urea (**blue**).

**Figure 4 pharmaceutics-13-00671-f004:**
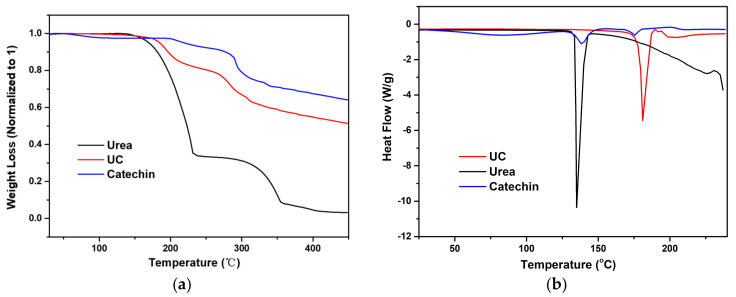
(**a**) TGA curves of urea, catechin and UC. (**b**) DSC curves of urea, catechin and UC.

**Figure 5 pharmaceutics-13-00671-f005:**
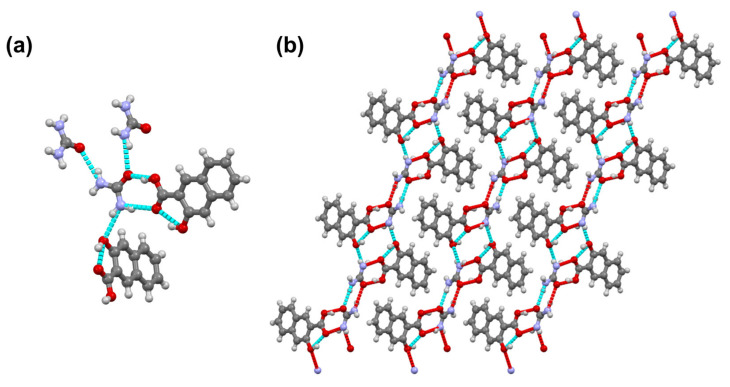
(**a**) Hydrogen bonds in UH. (**b**) View of crystal structure of UH along the b axis.

**Figure 6 pharmaceutics-13-00671-f006:**
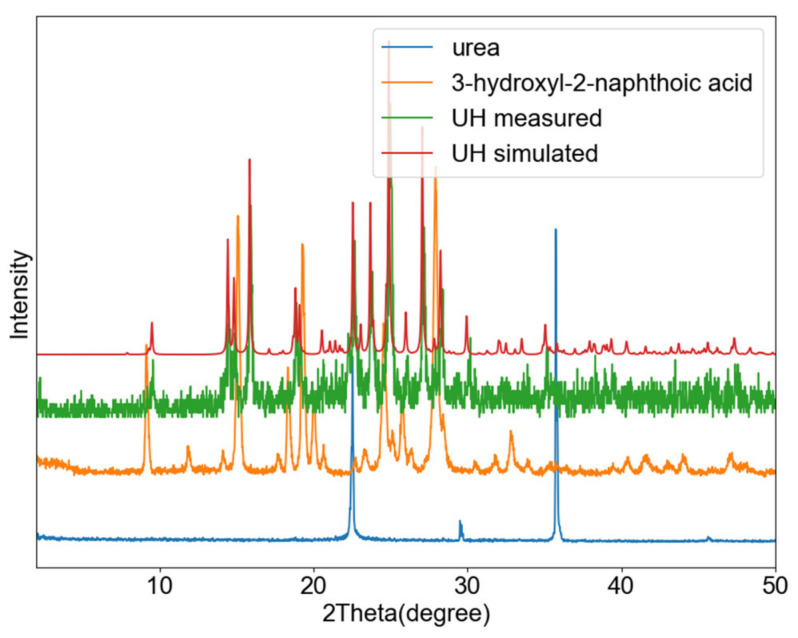
PXRD profiles of urea (**blue**), UH co-crystal obtained by grinding (**green**), the simulated UH pattern (**red**) and the experimental 3-hydroxyl-2-naphthoic acid pattern (**orange**).

**Figure 7 pharmaceutics-13-00671-f007:**
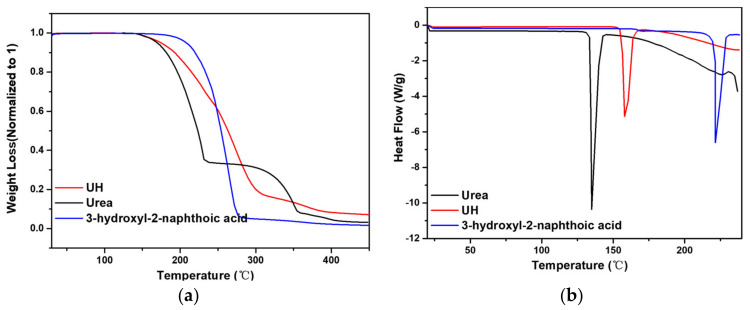
(**a**) TGA curves of urea, catechin and UC. (**b**) DSC curves of urea, catechin and UC.

**Figure 8 pharmaceutics-13-00671-f008:**
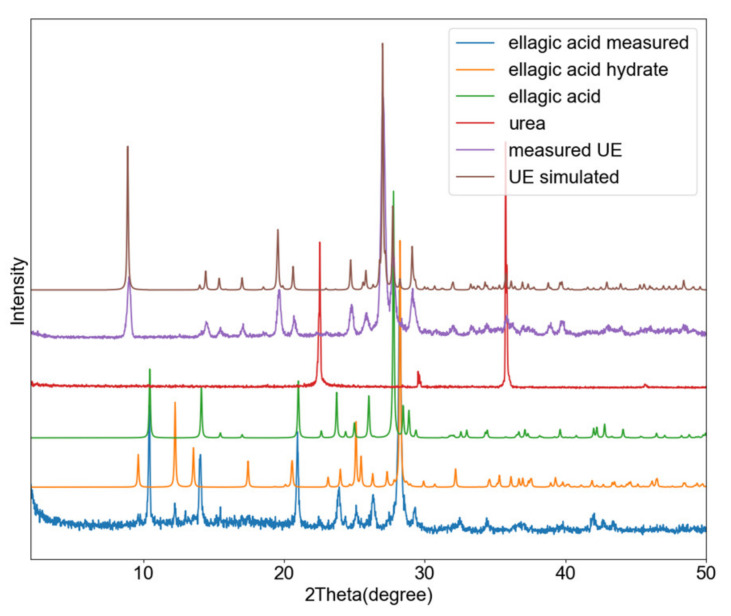
PXRD profiles of ellagic acid (**blue**), simulated ellagic acid hydrate (**orange**), simulated ellagic acid (**green**), urea (**red**), UE co-crystal obtained by grinding (**purple**), the simulated UH pattern (**brown**).

**Figure 9 pharmaceutics-13-00671-f009:**
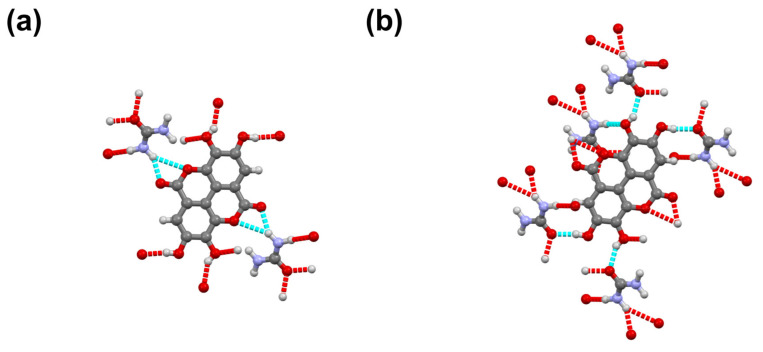
(**a**) Hydrogen bonds in UE between urea amide group and ellagic acid carbonyl group. (**b**) Hydrogen bonds formed by the phenol group in UE.

**Figure 10 pharmaceutics-13-00671-f010:**
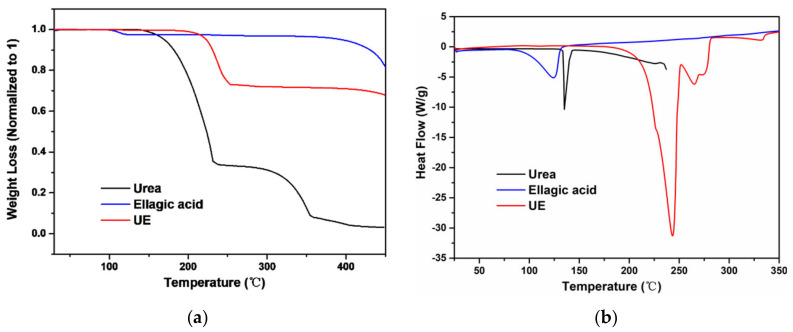
(**a**) TGA curves of urea, ellagic acid and UE. (**b**) DSC curves of urea, ellagic acid and UE.

**Figure 11 pharmaceutics-13-00671-f011:**
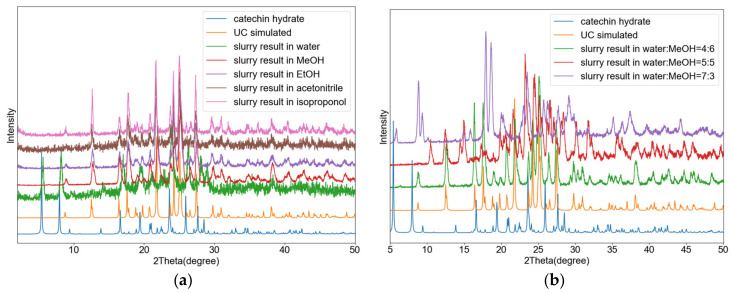
(**a**) PXRD profiles of congruence experiments results of UC in different solvents. (**b**) Various products obtained using different methanol/water ratio when suspending stoichiometric ratio of urea and catechin.

**Table 1 pharmaceutics-13-00671-t001:** SC-XRD crystallographic data for the UC and UH cocrystals.

Compound	UC Cocrystal	UH Cocrystal
Formula	C_16_H_18_N_2_O_7_	C_12_H_12_N_2_O_4_
*D_calc._*/g cm^−3^	1.544	1.398
*m*/mm^−1^	0.123	0.107
Formula Weight	350.32	248.24
Colour	Brown	colourless
Shape	needle	rod
Size/mm^3^	0.35 × 0.02 × 0.02	0.30 × 0.10 × 0.05
*T*/K	150(2)	293(2)
Crystal System	monoclinic	monoclinic
Space Group	*P*2_1_	*C*2/*c*
*a*/Å	10.7771(12)	24.353(2)
*b*/Å	5.0024(5)	5.0996(4)
*c*/Å	14.960(3)	20.7056(19)
*α*/^°^	90	90
*β*/^°^	110.849(17)	113.490(11)
*γ*/^°^	90	90
V/Å^3^	753.68(19)	2358.3(4)
*Z*	2	8
*Z’*	1	1
Wavelength/Å	0.71073	0.71073
Radiation type	MoK*_α_*	MoK*_α_*
Measured Refl’s.	3867	8859
Indep’t Refl’s	2127	2341
Refl’s I ≥ 2 *s*(I)	1217	1938
*R* _int_	0.1191	0.0365
GooF	1.026	1.063
*wR*_2_ (all data)	0.1510	0.1192
*wR_2_*	0.1240	0.1125
*R*_1_ (all data)	0.1472	0.0511
*R_1_*	0.0775	0.0421

## Data Availability

Not applicable.

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
