# Peer review of "Urea as a Cocrystal Former—Study of 3 Urea Based Pharmaceutical Cocrystals"

_pharmaceutics, 2021, doi:10.3390/pharmaceutics13050671_

Round 1

Reviewer 1 Report

Well done. No comments.

Author Response

Thank you for your positive review

Reviewer 2 Report

Formal remarks:

The authors used the journal template for editing the manuscript and forgot to overwrite or delete the unnecessary template text (e.g., a legend of Table 1 refers to Table 2 from the template without indicating the content, or below the References, the authors should delete the text that remained from the authors' instruction template).

Conceptual remarks:

The authors should discuss the connection between the solubility differences and the structural characteristics of the cocrystals. How does the long-range molecular order modify the solubility behavior? 

Author Response

We would like to thank the reviewer for his positive comments.
We removed all template text we forgot to remove (with exception of those the journal needs to fill in ).

We also commented upon the solubility behavior, but linking this directly to the structure only is rather impossible. We do agree that we could have specified and added following text:

'Solubility of a cocrystal depends on the free energy of the novel cocrystal as well as the solution free energy of dissolved compounds and their solution interaction. Predicting this solubility merely on the structure is not feasible. The increase in solubility for ellagic acid is not surprising as the solubility of ellagic acid is extremely low. Very likely a variation of free energy of the solid structure as well as a positive interaction between both components in solution needs to be taken into account.'

Reviewer 3 Report

The paper reports the synthesis, by mechanochemistry, of novel pharmaceutical cocrystals of catechin, 3-hydroxyl-2-naphthoic and ellagic acid with urea, a GRAS coformer.  The solids were characterized by differential scanning calorimetry, thermogravimetric analysis, powder X-ray diffraction and single crystal X-ray diffraction. Solubility measurements were also undertaken.

The paper is very well written, with clear structure and careful explanations throughout, enabling others to replicate these techniques if desired. The quality of experimental data is convincing and the conclusions appear to be reliable. I have just a comment, which the authors may wish to address:

The final goal of the synthesis of cocrystals, and in the particular case of pharmaceutical crystals, is to improve solubility, and it is of great importance to understand why a given coformer in some cases leads to an increase and in others to a decrease in solubility. Have the authors any explanation for the huge solubility enhancement of the ellagic acid/urea cocrystal relatively to pure ellagic acid and the decrease in solubility of catechin/urea and 3-hydroxyl-2-naphthoic/urea cocrystals comparatively with the pure compounds?

Author Response

We thank the reviewer for the positive review. His suggestion overlaps with that of reviewer 2. So we indeed clarified and gave a plausible suggestion as to why the solubility of ellagic acid is increased substantially. Following text was added:

'Solubility of a cocrystal depends on the free energy of the novel cocrystal as well as the solution free energy of dissolved compounds and their solution interaction. Predicting this solubility merely on the structure is not feasible. The increase in solubility for ellagic acid is not surprising as the solubility of ellagic acid is extremely low. Very likely a variation of free energy of the solid structure as well as a positive interaction between both components in solution needs to be taken into account.'